# Not All Frequencies Are Created Equal: Towards a Dynamic Fusion of Frequencies in Time-Series Forecasting

## ABSTRACT

Long-term time series forecasting is a long-standing challenge in various applications. A central issue in time series forecasting is that methods should expressively capture long-term dependency. Furthermore, time series forecasting methods should be flexible when applied to different scenarios. Although Fourier analysis offers an alternative to effectively capture reusable and periodic patterns to achieve long-term forecasting in different scenarios, existing methods often assume high-frequency components represent noise and should be discarded in time series forecasting. However, we conduct a series of motivation experiments and discover that the role of certain frequencies varies depending on the scenarios. In some scenarios, removing high-frequency components from the original time series can improve the forecasting performance, while in others scenarios, removing them is harmful to forecasting performance. Therefore, it is necessary to treat the frequencies differently according to specific scenarios. To achieve this, we first reformulate the time series forecasting problem as learning a transfer function of each frequency in the Fourier domain. Further, we design Frequency Dynamic Fusion (FreDF), which individually predicts each Fourier component, and dynamically fuses the output of different frequencies. Moreover, we provide a novel insight into the generalization ability of time series forecasting and propose the generalization bound of time series forecasting. Then we prove FreDF has a lower bound, indicating that FreDF has better generalization ability. Extensive experiments conducted on multiple benchmark datasets and ablation studies demonstrate the effectiveness of FreDF.

## CCS CONCEPTS

• **Computing methodologies** → Spectral methods; Time series forecasting;

## KEYWORDS

Time series forecasting, Fourier analysis, Dynamic fusion, Generalization analysis

## 1 INTRODUCTION

Time series forecasting is a well-established problem in various fields including energy usage [4], economic planning [1], weather

**Unpublished working draft. Not for distribution.**

alerts [10], and traffic forecasting [20]. With the development of deep learning [17], numerous methods have emerged for this forecasting tasks [2, 13, 35, 46]. A central issue in time series forecasting is that existing methods could not expressively capture long-term dependency, which is often characterized as periodicity and trends [5, 7, 11, 18, 36]. However, Fourier analysis has the strong potential to deal with long-term dependency, thereby makes related methods more flexible when adapted to different scenarios [37].

In the realm of time series forecasting, an effective approach to addressing long-term dependency is to utilize Fourier analysis [24, 37, 38, 44, 45]. Fourier analysis is a powerful method that represents complex time series as a series of cosine functions, each with its unique frequency [6]. This capability to represent infinitely long-term trends with a finite set of frequency components makes it efficiency when applied to long-term time series forecasting.

Existing methods based on Fourier analysis often assume that high-frequency components represent noise and should be discarded during forecasting tasks [44]. However, we argue that the role of certain frequencies varies in different scenarios. To validate this assumption, we conduct experiments on three datasets, eliminating low, middle, and high-frequency components respectively from the input of the training set to train a vanilla Transformer [34]. The results, depicted in Figure 1, suggest that eliminating certain frequencies may improve performance in specific datasets while decreasing in others. In Exchange-rate(Figure 1(e)), we get more accurate prediction results after eliminating high frequencies. But it is less precise in Figure 1(b). The same phenomenon occurs at other frequencies. More detailed experimental setup and analysis are provided in Section 3.

These findings emphasize that simply marking high-frequency components as noise is undesirable. Without prior knowledge, determining which frequencies compose noise remains uncertain [11]. Consequently, it is necessary to utilize different frequencies for forecasting and assign more rational weights to these forecasting results to improve the final prediction.

To separately assess the impact of different frequencies, it is necessary to predict each frequency individually. To begin with, we propose a mathematical reformulation of the time series forecasting task in the Fourier domain. Then we propose Frequency Dynamic Fusion (FreDF), a novel framework to process time series datasets in decomposition, forecasting, and dynamic fusion, which individually forecasts each Fourier component, and dynamically fuses the output of different frequencies. The advantage of dynamic fusion lies in its capacity to flexibly adjust the weights of each frequency component, leading to more precise predictions. Additionally, we propose the generalization bound of time series forecasting based on Rademacher complexity [3], and we prove that dynamic fusion improves the model's generalization ability. Experimental results on long-term forecasting datasets also confirm the superiority of FreDF.

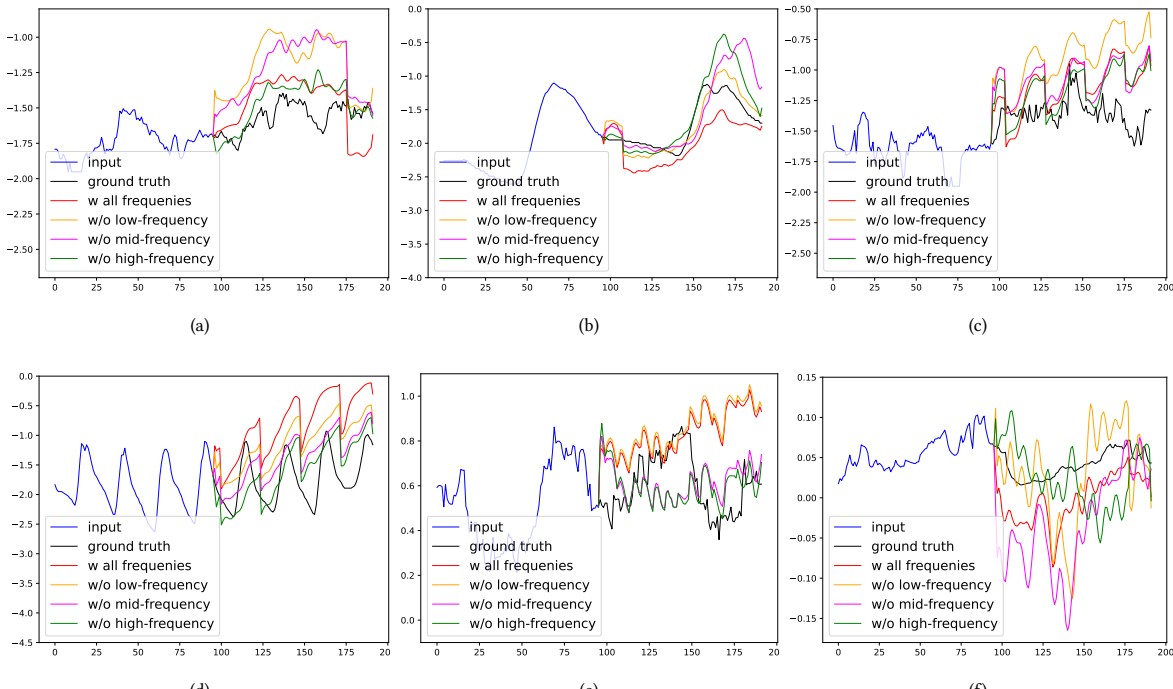

**Figure 1: The comparison of prediction results using different frequencies on different datasets. (a) ETTm1; (b)ETTm2; (c)ETTh1; (d)ETTh2; (e) Exchange-rate; (f) Weather.**

Overall, our contributions can be summarized as the following four points:

- We conduct a series of experiments to explore the role of different frequencies in prediction. Based on experimental phenomena we discover that the role of certain frequencies varies depending on the scenarios.
- We reformulate the time series forecasting problem as learning a transfer function in the Fourier domain. Further, we design FreDF, which individually forecasts each Fourier component, and dynamically fuses the output of different frequencies.
- We propose the generalization bound of time series forecasting. Then we prove FreDF has a lower bound, indicating that FreDF has better generalization ability.
- Extensive experiments conducted on various benchmark datasets demonstrate the effectiveness of FreDF.

## 2 RELATED WORK

With the advancement of deep learning, various methods, including CNN [2, 41], RNN [12, 33], and Transformer-based approaches [34], have been developed for time series forecasting tasks. While most previous works focus on learning models in the time domain (e.g., Informer [19], PeriodFormer [21], GCformer [43], Preformer [9], and Infomaxformer [32]), the core of these methods lies in utilizing correlations in the time domain to forecast future data.

In the Fourier domain, FEDformer [45] applies Transformer using Frequency Enhanced Blocks and Attention modules, and CoST

[37] explores learning seasonal representations. FEDformer and TimesNet [38] utilize frequency for analysis and period calculation, mapping one-dimensional time series to two-dimensional. FiLM [44] retains low-frequency Fourier components. However, these methods, involving Fourier analysis, do not explicitly model time series forecasting problems in the Fourier domain. In contrast, we reformulate the time series forecasting problem as learning a transfer function of each frequency in the Fourier domain.

Classical time series decomposition techniques [5] have been utilized to decompose time series into seasonal and trend components for interpretability. For instance, Autoformer [39] decomposes the data into trend and seasonal components, then employs the Transformer architecture for independent forecasts. Similarly, CoST [37] decomposes sequences into trend and seasonal components, carrying out separate forecasts in both time and Fourier domains. Different from these methods, our approach introduces a novel framework for dynamic decomposition, prediction, and fusion of time series.

## 3 EMPIRICAL ANALYSIS

Several studies suggest that high-frequency signals often represent noise and therefore should be discarded [44]. However, we argue that the role of certain frequencies is not universal and can be varied across different scenarios. In some cases, high-frequency signals may indeed be noise, while in others, they may hold valuable information. To confirm this idea, we conduct experiments on three datasets. The experimental settings and analysis are detailed below.

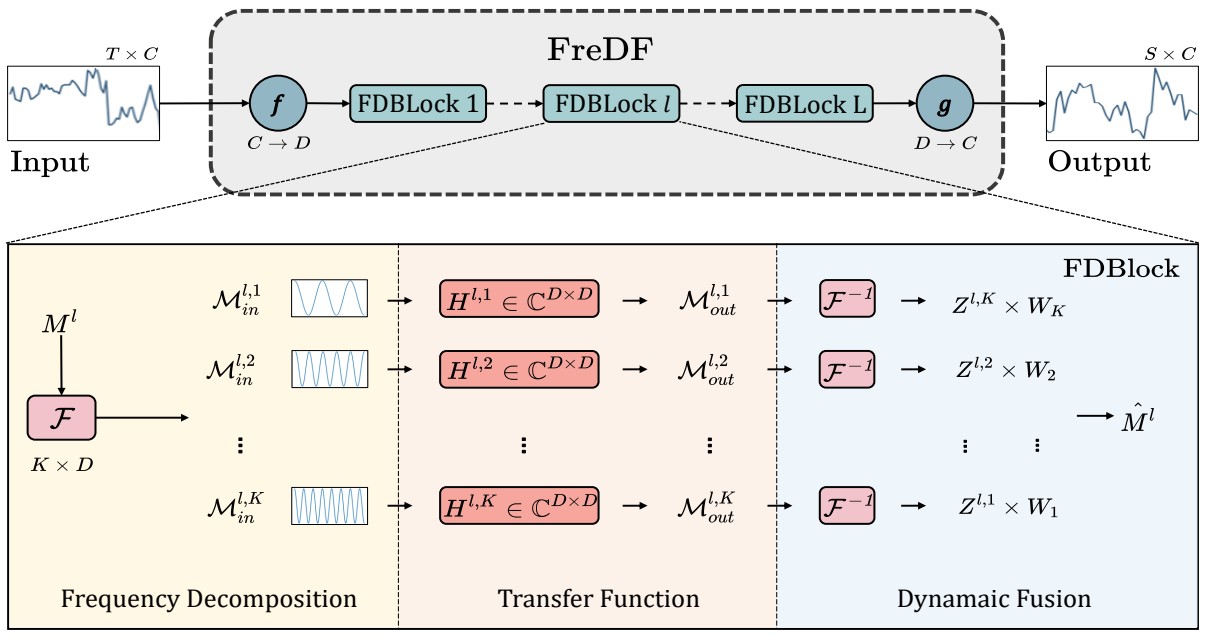

**Figure 2: Overall structure of FreDF, which consists of an Embedding module $f$ for embedding the feature dimension $C$ to $D$, a Projection module $g$ for projection back to $C$, and $L$ FDBlocks. In FDBlock, we decompose and forecast in the frequency domain, and dynamically fuse the prediction results for each Fourier component.**

## 3.1 Experimental Setup

We conduct experiments on six datasets: ETT(ETTm1, ETTm2, ETTh1, ETTh2), Weather, ECL, and Exchange-rate. For each dataset, we conduct a set of four forecasting tasks with the lookback length and prediction length both fixed to 96. The first task is the regular forecasting task. For the other three tasks, we transform the input series from the training set to the Fourier domain using Fast Fourier Transform (FFT) [28] and divide the frequency spectra into three subsets: the first third of the spectrum as low-frequency, the second third as middle-frequency, and the final third as high-frequency. We randomly set the Fourier coefficients corresponding to different subsets of the frequency spectrum to zero respectively in different experiments, and convert it back into the time domain as the input series. This step is to eliminate the influence of a certain subset of frequencies when training the model. We train an individual vanilla Transformer [34] following the standard setting [23] for each task in all three datasets. We visualize the prediction and ground truth of future series for all tasks and all datasets in Figure 1.

## 3.2 Experimental Observations and Analysis

Figure 1(a) shows that in the ETTm1 dataset, after eliminating high-frequency signals, the prediction results are closer to the ground truth compared to using all frequencies. However, the prediction results are further from the ground truth after eliminating low-frequency or mid-frequency signals. Figure 1(b) shows that in the ETTm2 dataset, we will get more accurate prediction results after eliminating low-frequency from historical time series. On the contrary, we will get worse prediction results after eliminating low-frequency form historical time series, shown in Figure 1(c). In Figure 1(d), no matter which subsets of the frequency we eliminate in the ETTh2 dataset, the prediction results are more accurate than those obtained using the original frequency for prediction, among which, eliminating high frequency has a better effect. As shown in Figure 1(e), in the Exchange-rate dataset, the prediction results are more accurate in the long term after eliminating mid-frequency or high-frequency signals. Conversely, the results predicted by eliminating low-frequency signals or using all signals are closer to ground truth in the mid-term. In the weather dataset, which is shown in Figure 1(f), the results predicted by eliminating low-frequency signals are more accurate in the short-term, predicted by eliminating high-frequency are more accurate in the mid-term, and predicted by eliminating mid-frequency are more accurate in the long term.

Yet, with the absence of prior knowledge, it remains challenging to distinguish noise from vital features. Therefore, we cannot merely mark high-frequency signals as noise. Considering this, it is necessary to utilize different frequencies for forecasting and subsequently adopt a more rational method to weight these forecasting results, thus attaining the final prediction.

## 4 METHOD

In this section, we begin by reformulating the time series forecasting problem in the Fourier domain. Subsequently, we propose FreDF, a model designed to predict the output of each frequency component

respectively, then combine each output of different components using a dynamic fusion strategy. We also present theoretical evidence supporting the idea that this dynamic fusion strategy enhances the generalization ability of FreDF.

## 4.1 Time series forecasting in Fourier domain

To achieve effective long-term forecasting, the model must go beyond merely memorizing past data points; it needs to grasp the underlying physical rules or inherent dynamics of the observed phenomena [26]. These dynamics governing the behavior of the time series, are presumed to be independent and unchanging over time [30]. In Fourier analysis, any time series can be represented by a set of orthogonal bases, i.e., the Fourier components; this orthogonal characteristic helps represent each rule with the dynamic of a single Fourier component [16]. In this section, we assume that the time series forecasting task is under a Linear Time-invariant (LTI) condition for the independent and time-invariant property of the inherent dynamics without loss of generality.

Specifically, from [30], let $x(t) \in \mathcal{I}$ be the input function and $y(t) \in O$ be the output function, they are both functions of time $t$ defined in Banach space $\mathcal{I}$ and $O$. The output of the LTI system can be defined as:

$$y(t) = \int_0^\infty h(t - \tau)x(\tau)d\tau. \qquad (1)$$

The goal of time-series forecasting can be regarded as finding a suitable transfer function $h : \mathcal{I} \to O$.

In discrete case, the Equation 1 can be express as:

$$Y[n] = h[n] * X = \sum_{m=0}^\infty h[n - m]X[m], \qquad (2)$$

here, $X[n]$ and $Y[n]$ is the discrete form of $x(t)$ and $y(t)$, respectively, $n \in [0, 1, \ldots, N]$, $N$ is the length of time series, and $*$ is the convolution operator. The output series $Y = [Y[0], Y[1], \ldots, Y[N]]$ are obtained by applying the convolution operator between $h$ and $X$.

The Discrete Fourier Transform (DFT) $\mathcal{F}$ [29] can transform $X$ from a function of discrete time to a function of Fourier component $k$:

$$\mathcal{X}[k] = \mathcal{F}(X)[k] = \sum_{n=0}^{N-1} X[n] \cdot e^{-j\frac{2\pi}{N}kn}, \qquad (3)$$

where $j$ is the imaginary unit, $\mathcal{X}[k]$ is the $k$-th Fourier components, $k \in [0, 1, \ldots, K]$ and $K$ is the total number of Fourier components.

THEOREM 4.1. (The convolution theorem [14]). The convolution theorem states that the Fourier transform of a convolution of two functions equals the point-wise product of their Fourier transform:

$$\mathcal{F}(h * X)[k] = \mathcal{F}(h)[k] \cdot \mathcal{F}(X)[k]. \qquad (4)$$

Applying DFT to the output sequence $Y$ according to Theorem 4.1 can convert the convolution in Equation 2 into a multiplication in the Fourier domain as:

$$\mathcal{Y}[k] = \mathcal{F}(h * X)[k] = \mathcal{F}(h)[k] \cdot \mathcal{X}[k]. \qquad (5)$$

Note that $h$ is an unknown operator in the aforementioned analysis. Therefore we propose to estimate $\mathcal{F}(h)$ directly with a learnable matrix $H_\theta$, where $\theta$ is the parameter. The transfer process is:

$$\hat{\mathcal{Y}}[k] = H_\theta \cdot \mathcal{X}[k], \qquad (6)$$

where $\hat{\mathcal{Y}}$ is the estimated output in Fourier domain.

Applying inverse Discrete Fourier transform $\mathcal{F}^{-1}$ (iDFT) can convert the estimated output back to the time domain with:

$$\hat{Y}[n] = \mathcal{F}^{-1}(\hat{\mathcal{Y}}) = \frac{1}{N} \sum_{k=1}^K \hat{\mathcal{Y}}[k] \cdot e^{j\frac{2\pi}{N}kn}. \qquad (7)$$

The learning objective for the learnable matrix is then to minimize the Mean Square Error (MSE) between the estimated output and the ground truth of the output:

$$\min_\theta \frac{1}{N} \sum_{n=0}^{N-1} (Y[n] - \hat{Y}[n])^2. \qquad (8)$$

So far, the time series forecasting problem in the time domain has been reformulated as learning a transfer function $H_\theta$ in the Fourier domain.

## 4.2 Frequency Dynamic Fusion

Based on the findings in Section 3, there is no universal criteria to determine the importance of a specific frequency in different situations, for the role of certain frequency changes across various scenarios. For instance, a frequency may be crucial in one scenario but negatively impact performance in another. To address this variability, we propose FreDF (Frequency Dynamic Fusion), which dynamically calculates the weights for the estimated prediction of each frequency, taking their importance into account. Our proposed FreDF consists of the Embedding, the FDBlock, and the Projection layers. We provide the pseudo-code of FreDF in algorithm 1.

To predict the future $S$ timestamps, we padding $X[n]$ in time dimension with $S$ zeros as unknown data.

*4.2.1 Embedding.* In the Embedding module, we lift the input time series into an embedding space:

$$M^1[n] = f(X[n]), \qquad (9)$$

here, $M^1[n]$ is the embedded representation of the input time series, $f : \mathbb{R}^C \to \mathbb{R}^D$ is a multi-layer perceptron (MLP) used for the embedding, $C$ is the number of variables in the input time series, and $D$ is the dimension of the embedding space. It's crucial to note that we are embedding the feature dimensions, not the time dimensions. This means that the transformation does not affect the temporal characteristics of the data. Therefore, subsequent operations, such as Fourier transformations that target the time dimensions, remain unaffected by the embedding process.

*4.2.2 FDBlocks.* Within each FDBlock, we first apply Fast Fourier Transform (FFT)[28] to the input embedding $M^l[n]$, transforming it into the Fourier domain, which is an efficient algorithm to perform DFT:

$$\mathcal{M}^l[k] = \mathcal{F}(M^l)[k], \qquad (10)$$

here $\mathcal{M}^l[k]$ is the Fourier components, $l = 1, 2, \ldots, L$ denotes the $l$-th FDBlock and $L$ is the total number of FDBlocks.

To facilitate this independent processing, we propose a decoupling strategy. Instead of treating the Fourier components $\mathcal{M}^l[k] \in$

**Algorithm 1** Pseudo-Code of FreDF

---

**Input:** Time series data $X \in \mathbb{R}^{T \times C}$, lookback length $T$, predict length $S$, variables number $C$, FDBlock number $L$, token dimension $D$, $K$ is computed as $K = \frac{T+S}{2} + 1$, frequency spectrum length $K$.

---

1: $X' = \text{ZeroPadTimeSeries}(X, S)$     ▷ $\mathbf{X} \in \mathbb{R}^{(T+S) \times C}$
2: $M^1 = \text{MLP}(X')$     ▷ $M^1 \in \mathbb{R}^{(T+S) \times D}$
3: **for** $l = 1$ to $L$ **do**
4:     **for** $m = 1$ to $K$ **do**
5:        $\mathcal{M}^l[k] = \mathcal{F}(M^l)[k]$     ▷ $\mathcal{M}^l \in \mathbb{R}^{K \times D}$
6:        **for** $k = 1$ to $K$ **do**
7:           **if** $k \neq m$ **then**
8:              $\mathcal{M}_{in}^{l,m}[k] = 0$
9:           **else**
10:              $\mathcal{M}_{in}^{l,m}[k] = \mathcal{M}^l[k]$
11:           **end if**
12:        **end for**
13:        Learn transfer function $H^{l,m}$     ▷ $H^{l,m} \in \mathbb{R}^{D \times D}$
14:        $\mathcal{M}_{out}^{l,m}[k] = \mathcal{M}_{in}^{l,m}[k] \cdot H^{l,m}$
15:        $Z^{l,m} = \mathcal{F}^{-1}(\mathcal{M}_{out}^{l,m})$
16:     **end for**
17:     $M^{l+1} = \hat{M}^l = \sum_{m=0}^{K} Z^{l,m} \cdot W_m,$     ▷ $\hat{M}^l \in \mathbb{R}^{(T+S) \times D}$
18: **end for**
19: $\hat{Y} = MLP(\hat{M}^L)[T : T + S, :]$
20: **return** $\hat{Y}$     ▷ Return the prediction results

---

$\mathbb{C}^{K \times D}$ as a whole, we create $K$ copies of each frequency component and only retain the $m$-th frequency in each copy, denoted as $\mathcal{M}_{in}^{l,m}[k]$:

$$\mathcal{M}_{in}^{l,m}(k) = \begin{cases} 0 & \text{if } k \neq m \\ \mathcal{M}^l(k) & \text{if } k = m \end{cases}, k = 0, 1, ..., K. \tag{11}$$

This strategy allows us to maintain the original dimensionality of the data while enabling independent processing of each frequency. Next, based on subsection 4.1, we aim to learn transfer functions $H^{l,m} \in \mathbb{C}^{D \times D}$, $m \in [1, \ldots, K]$ for each independent component $\mathcal{M}_{in}^{l,m} = \mathcal{M}^l[m]$, $m \in [1, \ldots, K]$, and obtain the estimated output $\mathcal{M}_{out}^{l,m}$ in the Fourier domain with:

$$\mathcal{M}_{out}^{l,m} = \mathcal{M}_{in}^{l,m} \cdot H^{l,m}. \tag{12}$$

The estimated output for frequency $m$ in the time domain $Z^{l,m}[n]$ can be obtained by applying inverse Fast Fourier Transform (iFFT) to $\mathcal{M}_{out}^{l,m}$. The result of this operation is represented as:

$$Z^{l,m}[n] = \mathcal{F}^{-1}(\mathcal{M}_{out}^{l,m})[n]. \tag{13}$$

So far we have decomposed the prediction process of each individual frequency $m$.

Next, we apply a trainable weight vector $W \in \mathbb{R}^K$, where each component $W_m$ represents the importance of the $m$-th frequency when predicting the output embedding. The estimated output $\hat{M}^l[n]$ is then represented as a weighted sum of all the individual frequency predictions $Z^{l,m}[n]$, with each prediction multiplied by its corresponding weight $W_m$. The estimated output $\hat{M}^l[n]$ is represented as

a weighted sum of all the individual frequency predictions $Z^{l,m}[n]$, as given by the following equation:

$$\hat{M}^l[n] = \sum_{m=0}^{K} Z^{l,m}[n] \cdot W_m, \tag{14}$$

where each prediction $Z^{l,m}[n]$ is multiplied by its corresponding weight $W_m$ and the $W_m$ can be either static or dynamic, i.e. fixed or learnable.

The FDBlock is formulated as an iterative architecture, where each output $\hat{M}^l[n]$ of the $l$-th layer serves as the input of the $(l+1)$-th layer.

During the training process, we aim to learn the transfer functions $H^{l,m} \in \mathbb{C}^{D \times D}$ and the weight vector $W \in \mathbb{R}^K$ by minimizing a loss function, which measures the difference between the estimated output $\hat{Y}[n]$ and the true output $Y[n]$.

*4.2.3 Projection.* After $L$ FDBlocks, we apply another MLP $g : \mathbb{R}^D \to \mathbb{R}^C$ to the final estimated output $\hat{M}^L[n]$, projecting it back to the variable space. The result of this operation is represented as:

$$\hat{Y}[n] = g(\hat{M}^L[n])[T : T + S, :]. \tag{15}$$

## 4.3 Theoretical Analysis

In this subsection, we provide a theoretical analysis to demonstrate the effectiveness of our dynamic fusion method. Without loss of generality, time series forecasting methods could be regarded as auto-regressive models [5], from this perspective, we indicate that the generalization ability of time series prediction models could be reflected in the following two aspects: the capacity to capture the long-term dependency of time series, as well as the capacity to achieve good prediction results in different scenarios.

For simplicity, consider the fusion strategy in a regression setting using a mean squared loss function. Firstly, we propose to characterize the generalization error bound using Rademacher complexity [3] and separate the bound into three components (Theorem 4.2). Meanwhile, we also give further proof based on the above separation to illustrate that the dynamic fusion method achieves a better ability to capture long-term dependency under certain conditions (Theorem 4.3). Secondly, we demonstrate that the quantity of parameters in our method is fewer than compared methods, which indicates that our method is more flexible to apply to more scenarios, experiment results in Section 5.3 also validate our illustration as well. See Appendix A for details.

Specifically, we use $\mathcal{X}$, $\mathcal{Y}$, and $\mathcal{Z}$ to denote the input space (historical sequence), target space (prediction sequence), and latent space. Define $u : \mathcal{X} \to \mathcal{Z}$ is a fusion mapping from the input space to the latent space, $g : \mathcal{Z} \to \mathcal{Y}$ is a task mapping. Our goal is to learn the fusion operator $f = g \circ u(x)$, which is essentially a regression model. Under an $H$ frequency components scenario, $f^h$ is the frequency-specific composite function of frequency component $x^h$. The final prediction of the dynamic fusion method is calculated by: $f(x) = \sum_{h=1}^{H} w^h \cdot f^h(x^h)$, where $f(x)$ denotes the final prediction. In contrast to static fusion, i.e., every frequency is given a predefined weight which is a constant, dynamic fusion calculates the weights of every frequency dynamically. To distinguish them, denotes $w_{static}^h$ the weight of frequency $h$ in static situation

and $w_{dynamic}^h$ the weight of frequency $h$ in dynamic situation. The generalization error of regression model $f$ is defined as:

$$\text{GError} = \mathbb{E}_{(x,y)\sim\mathcal{D}}[l(f(x),y)], \tag{16}$$

where $\mathcal{D}$ is the unknown joint distribution, $l$ is mean squared loss function. For convenience, we simplify the regression loss $l(f^h(x^h),y)$ as $l^h$. Now we present the first main result of frequency fusion.

THEOREM 4.2. Given the historical sequence $X_T \in \mathbb{R}^{T \times C}$ and the ground truth of prediction sequence $Y_{T'} \in \mathbb{R}^{T' \times C}$, $\hat{E}(f^h)$ is the empirical error of $f^h$ on frequency $h$. Then for any hypothesis $f$ in the finite set $F$ and $1 > \delta > 0$, with probability at least $1 - \delta$, it holds that

$$\text{GError} \leq \sum_{h=1}^{H} \mathbb{E}(w^h)\hat{E}(f^h) + \sum_{h=1}^{H} \mathbb{E}(w^h)\mathfrak{R}_h(f^h) \\ + \sum_{h=1}^{H} Cov(w^h,l^h) + M\sqrt{\frac{\ln(1/\delta)}{2H}}, \tag{17}$$

where $\mathbb{E}(w^h)$ represents the expectations of fusion weights on joint distribution $\mathcal{D}$, $\mathfrak{R}_h(f^h)$ represents Rademacher complexity, and $Cov(w^h,l^h)$ represents the covariance between fusion weight and loss.

Theorem 4.2 demonstrates that the generalization error of the regression model is bounded by the weighted average performances of all regression operators for each frequency in terms of empirical loss, model complexity, and the covariance between fusion weight and regression loss of all frequencies. After the general error bound is established, the next goal is to verify if dynamic fusion indeed achieves a tighter bound than that of static fusion. Informally, in Equation 17, the covariance term measures the joint variability of $w^h$ and $l^h$. However, in static fusion, $w_{static}^h$ is a constant, which means that the covariance is equal to zero for any static fusion method. Thus the generalization error bound of static fusion methods is reduced to:

$$\text{GError}(f_{static}) \leq \sum_{h=1}^{H} (w_{static}^h)\hat{E}(f^h) \tag{18} \\ + \sum_{h=1}^{H} (w_{static}^h)\mathfrak{R}_h(f^h) + M\sqrt{\frac{\ln(1/\delta)}{2H}}.$$

So when the summation of the average empirical loss, the average complexity is invariant or smaller in dynamic fusion and the covariance is no greater than zero, we can ensure that dynamic fusion provably outperforms static fusion. This theorem is formally presented as:

THEOREM 4.3. Let $\overline{\text{GError}}(f_{dynamic})$, $\overline{\text{GError}}(f_{static})$ be the upper bound of generalization regression error of dynamic and static fusion method respectively. $\hat{E}(f^h)$ is the empirical error defined in Theorem 4.2. Then for any hypothesis $f_{dynamic}$, $f_{static}$ in finite set $F$ and $1 > \delta > 0$, it holds that

$$\overline{\text{GError}}(f_{dynamic}) \leq \overline{\text{GError}}(f_{static}) \tag{19}$$

with probability at least $1 - \delta$, if we have

$$\mathbb{E}(w_{dynamic}^h) = w_{static}^h \tag{20}$$

and

$$r(w_{dynamic}^h, l^h) \leq 0 \tag{21}$$

for all frequencies, where $r$ is the Pearson correlation coefficient which measures the correlation between fusion weights $w_{dynamic}^h$ and the loss of each frequency $l^h$.

Theorem 4.2 and Theorem 4.3 verify that the dynamic fusion method has a lower generalization bound, which indicates the capacity to capture the long-term dependency of our method. Furthermore, suppose for each frequency, the regression operator used in dynamic and static fusion are of the same architecture, then the intrinsic complexity $\mathfrak{R}_h(f^h)$ can be invariant. Thus, in this case, it holds that

$$\sum_{h=1}^{H} \mathbb{E}(w_{dynamic}^h)\mathfrak{R}_h(f^h) \leq \sum_{h=1}^{H} (w_{static}^h)\mathfrak{R}_h(f^h). \tag{22}$$

In Equation 22, it is easy to derive the conclusion that our model has a lower average complexity, corresponding to a lower quantity of parameters during the training process. Experiment results in Section 5.3 also validate this conclusion.

## 5 EXPERIMENTS

In this section, we first provide the details of the implementation and datasets. Next, we present the comparison results on eight benchmark datasets. Lastly, we conduct ablation studies to evaluate the effectiveness of each module in our method.

### 5.1 Implement Details

All the experiments are implemented in PyTorch [31] and trained on NVIDIA V100 32GB GPUs. We use ADAM [15] with an initial learning rate in $\{10^{-3}, 10^{-4}\}$ and MSELoss for model optimization. An early stopping counter is employed to stop the training process after three epochs if no loss degradation on the valid set is observed. The mean square error (MSE) and mean absolute error (MAE) are used as metrics. All experiments are repeated 3 times and the mean of the metrics is used in the final results. The transfer function is implemented using the complex 64 data type in PyTorch. The batch size is set to 4 and the number of training epochs is set to 10. We set the number of FDBlocks in our proposed model $L \in \{1, 2, 3\}$. The dimension of series representations $D \in \{64, 128, 256, 512\}$, or it is not embedded at all. We set the dropout rates in $\{0, 0.2, 0.4\}$.

### 5.2 Main Results

We thoroughly evaluate the proposed FreDF on various long-term time series forecasting benchmarks. For better comparison, we follow the experiment settings of iTransformer in [23] the prediction lengths for both training and evaluation vary within the set $S \in \{96, 192, 336, 720\}$, with a fixed lookback length of $T = 96$.

5.2.1 *Baselines*. We carefully choose 10 well-acknowledged forecasting models as our benchmark, including (1) Transformer-based methods: iTransformer [23], Autoformer [39], FEDformer [45], Stationary [25], Crossformer [42], PatchTST [27]; (2) Linear-based

**Table 1: Long-term multivariate forecasting results with prediction lengths $S \in \{96, 192, 336, 720\}$ and fixed lookback length $T = 96$. The best Forecasting results in bold and the second underlined. The lower MSE/MAE indicates the more accurate prediction result.**

| Models | | FreDF (Ours) | | iTransformer [23] | | PatchTST [27] | | Crossformer [42] | | TiDE [8] | | TimesNet [38] | | DLinear [40] | | SCINet [22] | | FEDformer [45] | | Stationary [25] | | Autoformer [39] | |
|---|---|---|---|---|---|---|---|---|---|---|---|---|---|---|---|---|---|---|---|---|---|---|---|
| Metric | | MSE | MAE | MSE | MAE | MSE | MAE | MSE | MAE | MSE | MAE | MSE | MAE | MSE | MAE | MSE | MAE | MSE | MAE | MSE | MAE | MSE | MAE |
| ETTm1 | 96 | **0.324** | **0.367** | 0.334 | 0.368 | 0.329 | 0.367 | 0.404 | 0.426 | 0.364 | 0.387 | 0.338 | 0.375 | 0.345 | 0.372 | 0.418 | 0.438 | 0.379 | 0.419 | 0.386 | 0.398 | 0.505 | 0.475 |
| | 192 | **0.365** | 0.387 | 0.377 | 0.391 | 0.367 | **0.385** | 0.450 | 0.451 | 0.398 | 0.404 | 0.374 | 0.387 | 0.380 | 0.389 | 0.439 | 0.450 | 0.426 | 0.441 | 0.459 | 0.444 | 0.553 | 0.496 |
| | 336 | **0.391** | **0.405** | 0.426 | 0.420 | 0.399 | 0.410 | 0.532 | 0.515 | 0.428 | 0.425 | 0.410 | 0.411 | 0.413 | 0.413 | 0.490 | 0.485 | 0.445 | 0.459 | 0.495 | 0.464 | 0.621 | 0.537 |
| | 720 | 0.459 | **0.436** | 0.491 | 0.459 | **0.454** | 0.439 | 0.666 | 0.589 | 0.487 | 0.461 | 0.478 | 0.450 | 0.474 | 0.453 | 0.595 | 0.550 | 0.543 | 0.490 | 0.585 | 0.516 | 0.671 | 0.561 |
| | Avg | **0.384** | **0.398** | 0.407 | 0.410 | 0.387 | 0.400 | 0.513 | 0.496 | 0.419 | 0.419 | 0.400 | 0.406 | 0.403 | 0.407 | 0.485 | 0.481 | 0.448 | 0.452 | 0.481 | 0.456 | 0.588 | 0.517 |
| ETTm2 | 96 | **0.175** | **0.257** | 0.180 | 0.264 | 0.175 | 0.259 | 0.287 | 0.366 | 0.207 | 0.305 | 0.187 | 0.267 | 0.193 | 0.292 | 0.286 | 0.377 | 0.203 | 0.287 | 0.192 | 0.274 | 0.255 | 0.339 |
| | 192 | **0.241** | **0.299** | 0.250 | 0.309 | 0.241 | 0.302 | 0.414 | 0.492 | 0.290 | 0.364 | 0.249 | 0.309 | 0.284 | 0.362 | 0.399 | 0.445 | 0.269 | 0.328 | 0.280 | 0.339 | 0.281 | 0.340 |
| | 336 | **0.303** | **0.341** | 0.311 | 0.348 | 0.305 | 0.343 | 0.597 | 0.542 | 0.377 | 0.422 | 0.321 | 0.351 | 0.369 | 0.427 | 0.637 | 0.591 | 0.325 | 0.366 | 0.334 | 0.361 | 0.339 | 0.372 |
| | 720 | 0.405 | **0.396** | 0.412 | 0.407 | **0.402** | 0.400 | 1.730 | 1.042 | 0.558 | 0.524 | 0.408 | 0.403 | 0.554 | 0.522 | 0.960 | 0.735 | 0.421 | 0.415 | 0.417 | 0.413 | 0.433 | 0.432 |
| | Avg | **0.281** | **0.323** | 0.288 | 0.332 | 0.281 | 0.326 | 0.757 | 0.610 | 0.358 | 0.404 | 0.291 | 0.333 | 0.350 | 0.401 | 0.571 | 0.537 | 0.305 | 0.349 | 0.306 | 0.347 | 0.327 | 0.371 |
| ETTh1 | 96 | **0.367** | **0.397** | 0.386 | 0.405 | 0.414 | 0.419 | 0.423 | 0.448 | 0.479 | 0.464 | 0.384 | 0.402 | 0.386 | 0.400 | 0.654 | 0.599 | 0.376 | 0.419 | 0.513 | 0.491 | 0.449 | 0.459 |
| | 192 | **0.416** | **0.424** | 0.441 | 0.436 | 0.460 | 0.445 | 0.471 | 0.474 | 0.525 | 0.492 | 0.436 | 0.429 | 0.437 | 0.432 | 0.719 | 0.631 | 0.420 | 0.448 | 0.534 | 0.504 | 0.500 | 0.482 |
| | 336 | 0.477 | **0.443** | 0.487 | 0.458 | 0.501 | 0.466 | 0.570 | 0.546 | 0.565 | 0.515 | 0.491 | 0.469 | 0.481 | 0.459 | 0.778 | 0.659 | **0.459** | 0.465 | 0.588 | 0.535 | 0.521 | 0.496 |
| | 720 | **0.478** | **0.458** | 0.503 | 0.491 | 0.500 | 0.488 | 0.653 | 0.621 | 0.594 | 0.558 | 0.521 | 0.500 | 0.519 | 0.516 | 0.836 | 0.699 | 0.506 | 0.507 | 0.643 | 0.616 | 0.514 | 0.512 |
| | Avg | **0.435** | **0.431** | 0.454 | 0.447 | 0.469 | 0.454 | 0.529 | 0.522 | 0.541 | 0.507 | 0.458 | 0.450 | 0.456 | 0.452 | 0.747 | 0.647 | 0.440 | 0.460 | 0.570 | 0.537 | 0.496 | 0.487 |
| ETTh2 | 96 | **0.292** | **0.341** | 0.297 | 0.349 | 0.302 | 0.348 | 0.745 | 0.584 | 0.400 | 0.440 | 0.340 | 0.374 | 0.333 | 0.387 | 0.707 | 0.621 | 0.358 | 0.397 | 0.476 | 0.458 | 0.346 | 0.388 |
| | 192 | **0.376** | **0.391** | 0.380 | 0.400 | 0.388 | 0.400 | 0.877 | 0.656 | 0.528 | 0.509 | 0.402 | 0.414 | 0.477 | 0.476 | 0.860 | 0.689 | 0.429 | 0.439 | 0.512 | 0.493 | 0.456 | 0.452 |
| | 336 | **0.415** | **0.426** | 0.428 | 0.432 | 0.426 | 0.433 | 1.043 | 0.731 | 0.643 | 0.571 | 0.452 | 0.452 | 0.594 | 0.541 | 1.000 | 0.744 | 0.496 | 0.487 | 0.552 | 0.551 | 0.482 | 0.486 |
| | 720 | **0.420** | **0.439** | 0.427 | 0.445 | 0.431 | 0.446 | 1.104 | 0.763 | 0.874 | 0.679 | 0.462 | 0.468 | 0.831 | 0.657 | 1.249 | 0.838 | 0.463 | 0.474 | 0.562 | 0.560 | 0.515 | 0.511 |
| | Avg | **0.376** | **0.399** | 0.383 | 0.407 | 0.387 | 0.407 | 0.942 | 0.684 | 0.611 | 0.550 | 0.414 | 0.427 | 0.559 | 0.515 | 0.954 | 0.723 | 0.437 | 0.449 | 0.526 | 0.516 | 0.450 | 0.459 |
| Exchange | 96 | **0.082** | **0.199** | 0.086 | 0.206 | 0.088 | 0.205 | 0.256 | 0.367 | 0.094 | 0.218 | 0.107 | 0.234 | 0.088 | 0.218 | 0.267 | 0.396 | 0.148 | 0.278 | 0.111 | 0.237 | 0.197 | 0.323 |
| | 192 | **0.172** | **0.294** | 0.177 | 0.299 | 0.176 | 0.299 | 0.470 | 0.509 | 0.184 | 0.307 | 0.226 | 0.344 | 0.176 | 0.315 | 0.351 | 0.459 | 0.271 | 0.315 | 0.219 | 0.335 | 0.300 | 0.369 |
| | 336 | 0.316 | 0.405 | 0.331 | 0.417 | **0.301** | **0.397** | 1.268 | 0.883 | 0.349 | 0.431 | 0.367 | 0.448 | 0.313 | 0.427 | 1.324 | 0.853 | 0.460 | 0.427 | 0.421 | 0.476 | 0.509 | 0.524 |
| | 720 | **0.835** | **0.687** | 0.847 | 0.691 | 0.901 | 0.714 | 1.767 | 1.068 | 0.852 | 0.698 | 0.964 | 0.746 | 0.839 | 0.695 | 1.058 | 0.87 | 1.195 | 0.695 | 1.092 | 0.769 | 1.447 | 0.941 |
| | Avg | **0.351** | **0.396** | 0.360 | 0.403 | 0.367 | 0.404 | 0.940 | 0.707 | 0.370 | 0.413 | 0.416 | 0.443 | 0.354 | 0.414 | 0.750 | 0.626 | 0.519 | 0.429 | 0.461 | 0.454 | 0.613 | 0.539 |
| Weather | 96 | **0.157** | **0.208** | 0.174 | 0.214 | 0.177 | 0.218 | 0.158 | 0.230 | 0.202 | 0.261 | 0.172 | 0.220 | 0.196 | 0.255 | 0.221 | 0.306 | 0.217 | 0.296 | 0.173 | 0.223 | 0.266 | 0.336 |
| | 192 | **0.205** | **0.246** | 0.221 | 0.254 | 0.225 | 0.259 | 0.206 | 0.277 | 0.242 | 0.298 | 0.219 | 0.261 | 0.237 | 0.296 | 0.261 | 0.340 | 0.276 | 0.336 | 0.245 | 0.285 | 0.307 | 0.367 |
| | 336 | **0.259** | **0.287** | 0.278 | 0.296 | 0.278 | 0.297 | 0.272 | 0.335 | 0.287 | 0.335 | 0.280 | 0.306 | 0.283 | 0.335 | 0.309 | 0.378 | 0.339 | 0.380 | 0.321 | 0.338 | 0.359 | 0.395 |
| | 720 | 0.341 | **0.339** | 0.258 | 0.349 | 0.354 | 0.348 | 0.398 | 0.418 | 0.351 | 0.386 | 0.365 | 0.359 | 0.345 | 0.381 | 0.377 | 0.427 | 0.403 | 0.428 | 0.414 | 0.410 | 0.419 | 0.428 |
| | Avg | **0.241** | **0.270** | 0.258 | 0.279 | 0.259 | 0.281 | 0.259 | 0.315 | 0.271 | 0.320 | 0.259 | 0.287 | 0.265 | 0.317 | 0.292 | 0.363 | 0.309 | 0.360 | 0.288 | 0.314 | 0.338 | 0.382 |
| ECL | 96 | 0.150 | 0.242 | **0.148** | **0.240** | 0.181 | 0.270 | 0.219 | 0.314 | 0.237 | 0.329 | 0.168 | 0.272 | 0.197 | 0.282 | 0.247 | 0.345 | 0.193 | 0.308 | 0.169 | 0.273 | 0.201 | 0.317 |
| | 192 | **0.161** | **0.253** | 0.162 | 0.253 | 0.188 | 0.274 | 0.231 | 0.322 | 0.236 | 0.330 | 0.184 | 0.289 | 0.196 | 0.285 | 0.257 | 0.355 | 0.201 | 0.315 | 0.182 | 0.286 | 0.222 | 0.334 |
| | 336 | **0.176** | **0.268** | 0.178 | 0.269 | 0.204 | 0.293 | 0.246 | 0.337 | 0.249 | 0.344 | 0.198 | 0.300 | 0.209 | 0.301 | 0.269 | 0.369 | 0.214 | 0.329 | 0.200 | 0.304 | 0.231 | 0.338 |
| | 720 | **0.217** | **0.311** | 0.225 | 0.317 | 0.246 | 0.324 | 0.280 | 0.363 | 0.284 | 0.373 | 0.220 | 0.320 | 0.245 | 0.333 | 0.299 | 0.390 | 0.246 | 0.355 | 0.222 | 0.321 | 0.254 | 0.361 |
| | Avg | **0.176** | **0.268** | 0.178 | 0.270 | 0.205 | 0.290 | 0.244 | 0.334 | 0.251 | 0.344 | 0.192 | 0.295 | 0.212 | 0.300 | 0.268 | 0.365 | 0.214 | 0.327 | 0.193 | 0.296 | 0.227 | 0.338 |
| Solar-Energy | 96 | 0.214 | 0.247 | **0.203** | **0.237** | 0.234 | 0.286 | 0.310 | 0.331 | 0.312 | 0.399 | 0.250 | 0.292 | 0.290 | 0.378 | 0.237 | 0.344 | 0.242 | 0.342 | 0.215 | 0.249 | 0.884 | 0.711 |
| | 192 | **0.230** | **0.255** | 0.233 | 0.261 | 0.267 | 0.310 | 0.734 | 0.725 | 0.339 | 0.416 | 0.296 | 0.318 | 0.320 | 0.398 | 0.280 | 0.380 | 0.285 | 0.380 | 0.254 | 0.272 | 0.834 | 0.692 |
| | 336 | **0.242** | **0.266** | 0.248 | 0.273 | 0.290 | 0.315 | 0.750 | 0.735 | 0.368 | 0.430 | 0.319 | 0.330 | 0.353 | 0.415 | 0.304 | 0.389 | 0.282 | 0.376 | 0.290 | 0.296 | 0.941 | 0.723 |
| | 720 | **0.245** | **0.271** | 0.249 | 0.275 | 0.289 | 0.317 | 0.769 | 0.765 | 0.370 | 0.425 | 0.338 | 0.337 | 0.356 | 0.413 | 0.308 | 0.388 | 0.357 | 0.427 | 0.285 | 0.295 | 0.882 | 0.717 |
| | Avg | **0.232** | **0.259** | 0.233 | 0.262 | 0.270 | 0.307 | 0.641 | 0.639 | 0.347 | 0.417 | 0.301 | 0.319 | 0.330 | 0.401 | 0.282 | 0.375 | 0.291 | 0.381 | 0.261 | 0.381 | 0.885 | 0.711 |
| 1st Count | | **34** | **36** | 2 | 2 | 3 | 2 | 0 | 0 | 0 | 0 | 0 | 0 | 0 | 0 | 0 | 0 | 1 | 0 | 0 | 0 | 0 | 0 |

 

**Table 2: Ablation on the influence of transfer function.**

| Methods | Metric | ETTh1 | | | | ETTm1 | | | | Exchange-rate | | | |
|---|---|---|---|---|---|---|---|---|---|---|---|---|---|
| | | 96 | 192 | 336 | 720 | 96 | 192 | 336 | 720 | 96 | 192 | 336 | 720 |
| W Transfer function | MSE | **0.367** | **0.416** | **0.477** | **0.478** | **0.324** | **0.365** | **0.391** | **0.459** | **0.082** | **0.172** | **0.316** | **0.835** |
| | MAE | **0.397** | **0.424** | **0.443** | **0.458** | **0.367** | **0.387** | **0.405** | **0.436** | **0.199** | **0.294** | **0.405** | **0.687** |
| W/O Transfer function | MSE | 0.439 | 0.492 | 0.529 | 0.522 | 0.378 | 0.421 | 0.441 | 0.518 | 0.129 | 0.218 | 0.254 | 0.897 |
| | MAE | 0.444 | 0.505 | 0.561 | 0.541 | 0.405 | 0.432 | 0.439 | 0.487 | 0.251 | 0.344 | 0.312 | 0.709 |

**Table 3: Ablation between static fusion and dynamic fusion.**

| Models | | FreDF | | FreSF | |
|---|---|---|---|---|---|
| Metric | | MSE | MAE | MSE | MAE |
| Weather | 96 | **0.153** | **0.199** | 0.175 | 0.239 |
| | 192 | **0.205** | **0.246** | 0.215 | 0.276 |
| | 336 | **0.259** | **0.587** | 0.263 | 0.312 |
| | 720 | **0.341** | **0.339** | 0.343 | 0.377 |
| Exchange | 96 | **0.082** | **0.199** | 0.129 | 0.239 |
| | 192 | **0.172** | **0.294** | 0.231 | 0.332 |
| | 336 | **0.316** | **0.405** | 0.360 | 0.451 |
| | 720 | **0.835** | **0.687** | 0.891 | 0.741 |
| ETTh1 | 96 | **0.367** | **0.397** | 0.428 | 0.437 |
| | 192 | **0.416** | **0.424** | 0.475 | 0.456 |
| | 336 | **0.477** | **0.443** | 0.509 | 0.477 |
| | 720 | **0.478** | **0.458** | 0.509 | 0.490 |
| ETTh2 | 96 | **0.292** | **0.341** | 0.373 | 0.434 |
| | 192 | **0.376** | **0.391** | 0.441 | 0.462 |
| | 336 | **0.415** | **0.426** | 0.451 | 0.469 |
| | 720 | **0.420** | **0.439** | 0.459 | 0.480 |
| ETTm1 | 96 | **0.324** | **0.367** | 0.369 | 0.401 |
| | 192 | **0.365** | **0.387** | 0.419 | 0.430 |
| | 336 | **0.391** | **0.405** | 0.440 | 0.438 |
| | 720 | **0.459** | **0.436** | 0.497 | 0.468 |
| ETTm2 | 96 | **0.175** | **0.257** | 0.210 | 0.292 |
| | 192 | **0.241** | **0.299** | 0.279 | 0.337 |
| | 336 | **0.303** | **0.341** | 0.338 | 0.374 |
| | 720 | **0.405** | **0.396** | 0.449 | 0.436 |

**Table 4: Comparison of the number of parameters.**

| Models | Ours | iTransformer | PatchTST | FEDformer | FiLM |
|---|---|---|---|---|---|
| **params** | 151.4K | 3.1M | 3.5M | 14.0M | 12.0M |

methods: DLinear [40], TiDE [8]; and (3) TCN-based methods: SCINet [22], TimesNet [38].

*5.2.2 Forecasting Results.* Table 1 presents the results of FreDF in long-term multivariate forecasting with the best in **bold** and the second underlined. The lower MSE/MAE indicates the more accurate prediction result. Results demonstrate that our model performs optimally in 70 out of 80 benchmarks. Compared to FEDformer [45], FreDF shows an average improvement of 13% in terms of MSE and MAE, reaching up to 33% improvement on the Exchange-rate dataset. Compared to the best-performing Transformer-based model:iTransformer [23], FreDF consistently achieves superior performance across almost all datasets.

## 5.3 Ablation Study

In this section, we conduct ablation studies to examine the influence of transfer functions, dynamic fusion mechanisms, and the number of parameters in the proposed FreDF.

*5.3.1 Influence of transfer function.* We conduct an ablation study about the influence of the transfer function. We remove the transfer function in FreDF as the control group, follow the setup in Section 5.2, and carry out predictions on the ETTh1, ETTm1, and Exchange-rate dataset. We present the results in Table 2, which illustrates the crucial role of the transfer function and confirms the correctness of our analysis in Section 4.1.

*5.3.2 Influence of dynamic fusion.* We conduct an ablation study to investigate the influence of dynamic fusion. We replace the learnable weight vector with a fixed weight vector and name this modified model FreSF. Predictions are carried out on the ETT(ETTh1, ETTh2, ETTm1, ETTm2), Weather, and Exchang-rate datasets using the setup outlined in Section 5.2. The results are presented in Table 3. Additionally, we visualize the prediction results (with a prediction length $S = 96$) for both FreSF and FreDF in Appendix C. The experimental results demonstrate the effectiveness of the dynamic fusion strategy.

*5.3.3 Number of parameters.* We use iTransformer [23], patchTST [27], FEDformer [45] and FiLM [44] for comparison, and calculate the number of model parameters when forecasting the same task, present the results in Table 4. Our model despite using a relatively small number of parameters, can achieve good accuracy in prediction tasks. This also validates the superiority of our model, which is consistent with the theoretical analysis in Section 4.3.

## 6 CONCLUSION

In this paper, we experimentally explore the different roles of frequency in various scenarios. To better utilize these distinctions, we reformulate the problem of time series forecasting as learning transfer functions in the Fourier domain and design the FreDF model, which can independently forecast each Fourier component and dynamically fuse outputs from different frequencies. Then, we provide a novel understanding of the generalization ability of time series forecasting. Further, we also propose the generalization bound for time series forecasting and demonstrate that FreDF has a lower generalization bound, indicating its better generalization ability. Extensive experiments validate the effectiveness of FreDF on multiple benchmark datasets.

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
