# OpenReview forum: "Not All Frequencies Are Created Equal: Towards a Dynamic Fusion of Frequencies in Time-Series Forecasting"
_acmmm.org/ACMMM/2024/Conference — MM2024 Poster_

### Official Review · Reviewer_oZzd · 2024-05-19

**Rating:** 3
**Confidence:** 3

**Summary:**

To realize Time-Series Forecasting, this paper designs Frequency Dynamic Fusion (FreDF), which individually predicts each Fourier component, and dynamically fuses the output of different frequencies.

**Strengths:**

1. The idea of analyzing the effects of different frequency domains is interesting.

2. The performance of the model is proved by experiments.

3. Well written.

**Limitations:**

1. My biggest concern is that the current SOTA time series models mostly challege the setting that the input length is much smaller than the output prediction's length. In this paper, e.g. input: 96 vs. output: 720. In the main experiment, the history length is much smaller than the prediction length, i.e. T << S. A more practical setting is that for fixed prediction length S, the input length T should be viewed as a hyper-parameter to tune, which is used in [1,2]. Can you evaluate FreDF under this setting?

2. How many frequency domain components are used in this paper? How can authors obtain different frequency domain components after Fourier transform? From my current understanding, it seems that different components are obtained by downsampling. Some technical details need to be clearly stated.

3. I think the ablation experiments in different frequency domains in the supplementary material should be placed in the main text, because this is the core motivation of the author.

4. The main role of frequency domain analysis is to mine periodic and global patterns. Given that this is the core motivation of the authors, I recommend that they conduct experiments on datasets with stable periodicity [3], such as PEMS04 and PEMS08.

5. What are the advantages of the multi-frequency domain over the multi-patch or multi-granularity [4,5]?

[1]  Are Transformers Effective for Time Series Forecasting?

[2] A TIME SERIES IS WORTH 64 WORDS: LONG-TERM FORECASTING WITH TRANSFORMERS

[3] Exploring progress in multivariate time series forecasting: Comprehensive benchmarking and heterogeneity analysis

[4] Pathformer: Multi-scale Transformers with Adaptive Pathways for Time Series Forecasting

[5] TimeMixer: Decomposable Multiscale Mixing for Time Series Forecasting

**Suitability:**

2

---

### Official Review · Reviewer_DVSY · 2024-05-24

**Rating:** 5
**Confidence:** 3

**Summary:**

This paper proposes a method to learn a corresponding pattern for each frequency component. In my opinion, this is a new perspective.

**Strengths:**

1. The paper is well written, and easy to follow.

2. The paper proposes an interesting perspective to  explore the different roles of different frequency components.

2. The evaluation has demonstrated that the method have achieved superior performance.

**Limitations:**

1. Some frequency-based methods have not been discussed and compared, such as FITS[1] and FreTS[2].

2. The experiment part seems to be lack of the analysis of the transfer function (e.g., visualization) and efficiency analysis.

Questions:

1. Could you further provide more details about the definition, explanation of transfer function?

[1]: FITS: Modeling Time Series with 10k Parameters. ICLR 2024

[2]: Frequency-domain MLPs are More Effective Learners in Time Series Forecasting. NeurIPS 2023

**Suitability:**

2

---

### Official Review · Reviewer_ryBj · 2024-05-24

**Rating:** 3
**Confidence:** 3

**Summary:**

This paper analyzes how high-frequency components may play different roles in time series forecasting across various scenarios. It introduces Frequency Dynamic Fusion (FreDF), a method that individually predicts each Fourier component and dynamically integrates the outputs of different frequencies. Besides, they offer insight into the generalization capabilities of time series forecasting.

**Strengths:**

(1)  The paper introduces how the role of frequencies in forecasting changes depending on the scenario.

(2) They design a method called Frequency Dynamic Fusion (FreDF), which individually forecasts each Fourier component and dynamically fuses the outputs from different frequencies.

(3)  By proposing and validating a lower generalization bound for the FreDF method, this approach demonstrates enhanced generalization capabilities, ensuring robustness across various datasets and conditions.

**Limitations:**

(1) The paper focuses on time series prediction, specifically on frequency domain processing, which does not align closely with the main theme of the conference.

(2) The experimental design in the article arbitrarily segments the frequency domain into three stages based on their order of appearance, which lacks scientific rigor as frequencies typically overlap.

(3) The description of the dynamic fusion module's strategy in the article is somewhat vague and could benefit from clearer exposition.

**Suitability:**

1

---

### Official Review · Reviewer_e9Zd · 2024-05-26

**Rating:** 2
**Confidence:** 3

**Summary:**

This paper studies the frequency learning and analyze the useful frequencies in time series forecasting. Several experiments are conducted.

**Strengths:**

1. the writing is understandable.
2. the paper is easy to follow.

**Limitations:**

1. The experiments mostly compare with Transformer-based forecasting methods, but ignore the frequency-based time series forecasting methods, such as FreTS [1], FITS [2].
2. This paper didn't discuss many state-of-the-art frequency-based time series forecasting methods including [1-4], which makes the motivation and contribution lack novelty.

[1] Frequency-domain MLPs are More Effective Learners in Time Series Forecasting. In NeurIPS 2023.
[2] FITS: Modeling Time Series with 10⁢k Parameters. In ICLR 2024.
[3] FourierGNN: Rethinking multivariate time series forecasting from a pure graph perspective. In NeurIPS 2023.
[4] TSLANet: Rethinking Transformers for Time Series Representation Learning. In ICML 2024.

**Suitability:**

2

---

### Meta-Review · Area_Chair_hjeH · 2024-07-07

**Recommendation:** Accept (Poster)
**Confidence:** 3

**Metareview:**

The method tackles long-term time series forecasting challenges by emphasizing the capture of robust long-term dependencies and flexibility across scenarios. The authors introduce Frequency Dynamic Fusion (FreDF), predicting and integrating individual Fourier components dynamically. Additionally, the study explores generalization in time series forecasting and establishes a lower bound to demonstrate FreDF's effectiveness. Extensive experiments across diverse datasets validate the method's efficacy.

Initially, the reviewers are not supportive of the work (wr,br,wa,br), with main concerns on lacking comparision with SOTA, and lacking experiments with other settings. The rebuttal managed to address some concerns, where one reviewer raised the score from br to ba, two reviewers maintained the original score of br and wa, and one reviewer did not give the final review. The AC reviewed the paper and the rebuttal, and found the finding of the role of certain frequencies varies depending on the scenarios interesting, and the performance was good. It seems that, the request comparisions and experiments from reviewer e9Zd and oZzd are completed.